# Impact of β-defensin 103 (*DEFB103*) copy number variation on bull sperm parameters and post-insemination uterine gene expression

Ozge Sidekli[1], Edward J. Hollox[1], Sean Fair[2], Kieran G. Meade [3,4,5]*

1 Department of Genetics, Genomics and Cancer Sciences, University of Leicester, Leicester, United Kingdom, 2 Laboratory of Animal Reproduction, Department of Biological Sciences, Bernal Institute, Faculty of Science and Engineering, University of Limerick, Limerick, Ireland, 3 School of Agriculture and Food Science, University College Dublin, Dublin, Ireland, 4 UCD Conway Institute of Biomolecular and Biomedical Research, University College Dublin, Dublin, Ireland, 5 UCD Institute of Food and Health, University College Dublin, Dublin, Ireland

* kieran.meade@ucd.ie

## Abstract

Pregnancy rates for elite bulls used in artificial insemination (AI) can vary significantly and therefore the identification of molecular markers for fertility and targets to improve bull selection is important. β-defensin peptides have diverse regulatory roles in sperm function across multiple species but the role of copy number variation (CNV) on fertility parameters has not been previously evaluated. In this study, Holstein-Friesian bulls were screened based on reliable field fertility data to identify two groups (High and Low fertility (HF and LF, respectively)) of n = 10 bulls/group which were genotyped for β-defensin 103 (*DEFB103*) gene CNV by droplet digital PCR. Overall, low *DEFB103* copy number (CN) was associated with increased sperm motility across all bulls (n = 20, p < 0.05). As genetic diversity of *DEFB103* CN was only apparent in the LF group, three bulls per CNV class (low, intermediate and high CN) were chosen for more detailed comparative functional analysis. Sperm from low CN bulls exhibited higher binding to the oviductal epithelium, while high CN increased sperm membrane fluidity *in vitro* (p < 0.05). To investigate the functional effect of *DEFB103* CNV on the uterine response *in vivo*, 18 heifers were inseminated with sperm from bulls with low, intermediate and high CN. Transcriptomic analysis on uterine tissue harvested 12 h post-insemination showed differential expression of 58 genes (FDR < 0.1) involved in sperm migration, immune signalling and chemotaxis. Although field fertility results from a complex number of interactive factors, these novel results suggest a contributory role for *DEFB103* CN in both sperm function and the uterine response to bull sperm, thereby potentially contributing to pregnancy outcomes in cattle. Further analysis of the role of CNV in additional β-defensin genes in bull fertility is now warranted.

## Introduction

Fertility is critically important for the livestock industry, as it directly affects production outcomes as well as underpinning the economic sustainability of farming systems [1]. Increasing

**Data availability statement:** All RNA-seq data generated for this study have been deposited in the ArrayExpress database under the project accession number E-MTAB-14398. www.ebi.ac.uk/biostudies/arrayexpress/studies/E-MTAB-14398?

**Funding:** OS was funded by the Turkish Ministry of National Education, Republic of Turkiye postgraduate study abroad program. Support for travel for OS was provided in part by the Genetics Society. SF was funded by the Marie Skłodowska-Curie Doctoral Network, BullNet, grant number 101120104.

**Competing interests:** The authors have declared that no competing interests exist.

the efficiency of cattle production systems does not rely exclusively on female fertility but is also heavily influenced by that of the bull [2], particularly when artificial insemination (AI) is utilised on farm. This is especially true within a seasonal calving system, where a cow must produce a calf per year and calve early in early spring to maximise the amount of milk produced from low-cost grass forage. Historically, intense genetic selection for production traits has resulted in a reduction in fertility [3], and although selection traits have broadened in recent years, sub-optimal fertility remains one of the principal reasons for premature culling of cows [4]. Female fertility continues to be intensively researched but the role of the bull in determining successful pregnancy outcomes is often overlooked [2]. Despite extensive *in vitro* quality control checks of semen prior to its use in AI, individual bull's pregnancy rates can differ by up to 30% [5]. Various analytical techniques, such as computer-assisted sperm analysis (CASA) and flow cytometry, have been used to elucidate variation in semen quality, with consequences for bull fertility. With these methods, although 47% of the variation in bull fertility in the field can now be explained by *in vitro* evaluations [6,7], a large component of bull fertility remains unaccounted for. Given that a single bull used in AI can inseminate thousands of females per season, it is clear that optimal fertility of the sire is of critical importance to herd fertility outcomes.

The reasons for differences in fertility between AI bulls have been shown to be multifactorial, involving factors specific to the sperm developmental process prior to ejaculation as well as differential responses within the female tract. Research has identified explanatory factors related to season, ejaculate and bull age [8] and even molecular differences in methylation levels on spermatozoan DNA between bulls [9]. Post-ejaculation, differences in sperm transport and immune responses in the female tract [10,11] have been documented. In addition, the cryopreservation process itself [12] adds a potentially detrimental influence on sperm survival and physiology. Sperm are coated with an array of different multifunctional proteins, including a class of positively charged peptides known as β-defensins. β-defensins are classified by a conserved six-cysteine motif, which forms three disulphide bridges in a 1–5, 2–4, 3–6 arrangement [13]. Originally thought to provide protection with principal antimicrobial role, the pleiotropic peptides are now known to have extensive and important immunomodulatory properties [14]. In cattle, these molecules have not been as extensively studied as in other species but interestingly, cattle exhibit an expanded repertoire of β-defensin genes across 4 clusters in the bovine genome [15], one of which, on chromosome 13 shows preferential expression within the reproductive tract of the bull [16,17]. We have previously characterised the role of one of the members of this cluster - namely β-defensin 126 and demonstrated that the peptide is bound on sperm [18] and promotes sperm motility and sperm binding to oviductal epithelium [19,20]. In addition, a β-defensin haplotype has been found to be associated with fertility in bulls [21] indicating a potential role for additional β-defensin genes in the male fertility phenotype. This is of particular relevance as the deletion of a cluster of β-defensins when deleted in mice results in sperm dysfunction and sterility in males [22] and a variant of β-defensin 126 is associated with infertility and reduced sperm motility in men [23]. In cattle, subsequent research has almost exclusively focused on specific candidate β-defensins including β-defensin 126 showing the peptide is adsorbed on the tail and post-acrosomal region of the sperm [18] and promotes sperm motility and sperm binding to oviductal epithelium [19,20]. In addition, a β-defensin haplotype has been found to be associated with fertility in bulls used in AI [21]. Furthermore, recent research on the β-defensin gene family has identified recent gene duplications indicating expansion in certain livestock lineages with several mammalian lineages show evidence of multiple gene copies – known as copy number variation (CNV), in β-defensin genes between individuals [24]. This observation includes in cattle [25] and our recent work has identified that β-defensin 103 (*DEFB103*), in particular,

shows extensive multiallelic CNV and shows a diploid copy number (CN) range from 1 up to 29 gene copies across seven different cattle breeds [26]. This extensive number of additional copies of DEFB103 and variation between individual bulls suggests that they may play a determining role in fertility outcomes.

It is now thought that β-defensins play important reproductive roles while also protecting vulnerable sperm from the female immune system [17], making genes such as *DEFB103* relevant candidates for investigation into bull subfertility. Although shown to be CNV, functional analysis of *DEFB103* CN has not been previously performed and the role of these peptides on the bovine female immune response, post-ejaculation remains unstudied. Elucidating the full effect of all β-defensins contributing to fertility outcomes is on-going but it is anticipated that molecular markers will help improve diagnosis of infertility as well as identify genetic targets for enhanced breeding in livestock [27]. Given the extensive CNV shown in *DEFB103* [26], we hypothesize that *DEFB103* CNV is a strong genetic candidate for affecting bull fertility, with the encoded BBD103 protein potentially influencing fertility outcomes and related phenotypes. The objectives of this study were therefore to investigate the influence of *DEFB103* CNV on both bull sperm functional parameters, mucus penetration, and binding to the oviduct epithelium as well as post-insemination uterine gene expression from heifers inseminated with sperm from bulls representing low, intermediate and high *DEFB103* CNV groups.

## Materials and methods

### Ethical approval

The procedures were established in compliance with the Cruelty to Animals Act (Ireland 1876, as modified by European Communities regulations 2002 and 2005) and the European Community Directive 86/609/EC. Ethical approval was obtained from the University of Limerick Animal Ethics Committee.

### Reagents

Unless specified, all chemicals were obtained from Merck, Arklow, Co Wicklow, Ireland.

### Bull selection

Data on the field fertility of a population of Holstein-Friesian bulls (n = 840) used in Ireland were obtained from the Irish Cattle Breeding Federation (ICBF) database, selected so that each bull had a minimum of 500 inseminations based on an adjusted sire fertility (ASF) model [28]. ASF was defined as daughter pregnancy to a given service identified retrospectively either from a calving event or where a repeat service (or a pregnancy scan) deemed the animal not to be pregnant to the said service. Cows and heifers that were culled or died on farm were omitted. These raw data were then adjusted for factors including semen type (frozen, fresh), cow parity, days in milk, month of service, day of the week when serviced, service number, cow genotype, herd, AI technician, and bull breed and were weighted for number of service records resulting in an ASF rate centered at 0%. For this study, selected bulls from this fertility ranking were classified into either High Fertility (HF) with an average adjusted fertility score of + 6.5 or Low Fertility (LF), where the average was −6.6% (n = 10/ group). Genotyping data of bulls for *DEFB103* CNV was conducted as previously described using digital droplet PCR (ddPCR) [26]. For sperm functional analysis, 9 LF sires were further categorized based on their *DEFB103* CNV genotype (into three groups: low CN (n = 3 bulls (the lowest CN among the four bulls)), intermediate CN (n = 3 bulls), and high CN (n = 3 bulls)). We previously determined that the *DEFB103* genotype is multiallelic [26], and in this study a diploid CN of 6 was used as the reference, and distinct threshold values of < 4.5

and > 7.5 were used to define low CN and high CN of this gene, respectively. Among LF bulls with low, intermediate and high CNV, mean adjusted pregnancy rates were −8.96%, −3.96% and −7.8%, respectively (Table 1). The same degree of genetic variation in *DEFB103* CNV was not present in the selected HF bull group, with only low and intermediate CNV detected. This therefore precluded a more detailed comparative analysis of sperm function between CNV groupings.

Sperm physiology data were provided by Stiavnicka et al. [29], and were used to examine the effect of *DEFB103* CNV on various sperm motility and kinematic parameters (assessed using CASA), as well as functional traits including viability, membrane fluidity, and acrosomal status (assessed using flow cytometry) *in vitro* under both capacitating and non-capacitating conditions. These data were generated based on three straws per ejaculate (3 ejaculates per bull) in *in vitro* experiments. Modified Tyrode's medium (mTALP) was used to create the appropriate conditions for capacitation [30], which contains key ingredients such as NaHCO3, CaCl2, BSA and Heparin, and non-capacitation medium (NCM), is similar but lacks these ingredients, containing increased NaCl and polyvinyl alcohol. Frozen-thawed sperm were washed to remove the egg yolk-based extender through centrifugation at 300× g for 5 min and resuspended in 200 μL NCM, followed by a second centrifugation. The sperm

**Table 1. Individual-specific data on copy number variation (CNV) for the *DEFB103* gene across all three loci (including *DEFB103A*, *DEFB103A*-Like, and *DEFB103B*) and fertility phenotypes namely, low fertility (LF) and high fertility (HF) in Holstein Friesian bulls (n = 10 per group).**

| Individual bull ID | Fertility Status | Adjusted Sire Fertility (ASF) Rate (%) | Number of Inseminations which the ASF rate is based on | DEFB103 CN | DEFB103 CNV status |
|---|---|---|---|---|---|
| H1 | HF | 6.8 | 100288 | 2.22 | Low |
| H2 | HF | 6.5 | 3912 | 3.04 | Low |
| H3 | HF | 5.8 | 11459 | 2.76 | Low |
| H4 | HF | 6.8 | 12424 | 5.82 | Intermediate |
| H5 | HF | 7.1 | 34973 | 2.34 | Low |
| H6 | HF | 6.2 | 17441 | 3.09 | Low |
| H7 | HF | 6.7 | 1041 | 2.94 | Low |
| H8 | HF | 6.2 | 5119 | 4.38 | Low |
| H9 | HF | 6 | 8637 | 2.13 | Low |
| H10 | HF | 6.7 | 37849 | 5.82 | Intermediate |
| **Average ASF** | | **6.48 (SD 0.42)** | **Average CN** | **3.45** | |
| L1 | LF | −8.5 | 1034 | 2.19 | Low |
| L2 | LF | −4.3 | 23811 | 6.42 | Intermediate |
| L3 | LF | −3 | 568 | 6.93 | Intermediate |
| L4 | LF | −9.1 | 740 | 3.51 | Low |
| L5 | LF | −4.6 | 597 | 5.43 | Intermediate |
| L6 | LF | −3.5 | 1195 | 4.35 | Low |
| L7 | LF | −9.3 | 519 | 3.69 | Low |
| L8 | LF | −12.3 | 1477 | 13.05 | High |
| L9 | LF | −7.3 | 1772 | 19.35 | High |
| L10 | LF | −3.9 | 980 | 7.68 | High |
| **Average ASF** | | **−6.58 (SD 3.15)** | **Average CN** | **7.26** | |

Evaluating the combined loci of *DEFB103*, the classification of CNV for the diploid genome categorizes them as having low, intermediate, or high Copy Number (CN). Averages per group (±standard deviation for pregnancy rate) are shown.

pellet was then resuspended in either NCM or mTALP, with the concentration adjusted to 15 $\times 10^6$ sperm per mL. Sperm in mTALP were incubated at 38.5°C with 5% $CO_2$ for 4 h (representing capacitating conditions), while sperm in NCM were maintained at room temperature (representing non-capacitating conditions). To induce the acrosome reaction, Calcium Ionophore A23187 (CaI), along with a dimethyl sulfoxide (DMSO) vehicle control were used. A corresponding vehicle control with DMSO was included, and both samples were incubated at 37°C for 1 h.

## Ability of sperm from bulls with different *DEFB103* CNV genotypes to swim through artificial mucus

Sperm from two straws from each low fertility bull (n = 9) were diluted using a hemocytometer to attain a final sperm concentration of 10 x $10^6$ per mL in TALP medium, which included Hoechst 33342 fluorescent dye (10 mg/mL in 2.3% sodium citrate). This mixture was then incubated at 37°C for 5 min to remove semen extender. The ability of sperm to penetrate cervicovaginal mucus was assessed using the artificial mucus method previously described by Al Naib et al. [31]. Briefly, artificial mucus was formulated by diluting a sodium hyaluronate solution (MAP-5, Lab-stock MicroServices, Ireland) with phosphate-buffered saline, resulting in a final concentration of 6 mg of sodium hyaluronate per mL. Flattened capillary tubes (measuring 0.3 mm x 3.0 mm x 100 mm; Composite Metal Services Ltd, UK) were marked at 10 mm intervals from 10 to 90 mm. These tubes were subsequently filled with artificial mucus and sealed at one end. Two vertically positioned capillary tubes were placed in a 1.5 mL Eppendorf tube containing Hoechst-stained sperm. Thus, for each replicate, each bull sperm was represented by two capillary tubes. The tubes were incubated at 37°C for 1 h, then placed on a hotplate at 45°C for 1 min to immobilize the sperm. Sperm were counted across the width of the tube within one field of view wide, at 10 mm intervals using a fluorescent microscope (10x; Olympus BX 60).

## Ability of sperm from bulls with different *DEFB103* CNV genotypes to bind to oviductal explants *in vitro*

The *in vitro* oviductal epithelium binding capacity of sperm from three different ejaculates from low-fertility bulls, characterized by the *DEFB103* CNV genotype (low CN (n = 3), intermediate CN (n = 3) and high CN (n = 3)) was performed as per the method outlined by Lyons et al. [20]. Each bull's thawed semen was diluted with a prepared stock culture containing medium 199 (M199) supplemented with fetal bovine serum (10%) and gentamicin sulphate (0.25 mg/mL). The sperm concentration was adjusted to 5 x $10^6$ sperm per mL. Reproductive tracts were collected from non-pregnant heifers (n = 9) at a commercial abattoir. Various stages of the oestrous cycle were represented, as previous work by our group [20] and others [32] has indicated that the stage of the oestrous cycle does not impact sperm binding *in* vitro. Tissue explants were taken from the isthmic portion of the oviduct, and the explants from both oviducts were combined for each individual tract. Explants were prepared by cleaning oviducts, isolating the isthmic segment, and releasing epithelial cells through gentle compression and fragmentation. These were cultured in M199 medium supplemented with fetal calf serum and gentamicin sulphate, incubated to form everted vesicles, and used within 5 h of slaughter. For the binding assay, explants (20 μL) were combined with sperm (5 × $10^6$ sperm per mL) pre-stained with 1% Hoechst 33342 dye at 38.8°C for 30 min for enhanced binding visualisation [20]. After incubation and removal of loosely bound sperm by gently pipetting through two 75 μL droplets of M199 media on a heated 24-well culture plate, samples were examined under a fluorescent microscope. One drop (10 μL) of each treatment was placed

on a slide, a coverslip was added, and samples were imaged at 400× magnification using a microscope with a heated stage under half light and half fluorescence at 38.8°C. A total of ten explants from each treatment were randomly evaluated, and the density of sperm binding determined by calculating the number of bound spermatozoa per 0.1 mm² of explant surface. The assessor was blinded to the treatment for all sperm binding assessments.

### *In vivo* sperm migration into the oviduct and uterine transcriptomic response in bulls with different *DEFB103* CNV genotypes

**Heifer selection.** Eighteen 24- to 30-month-old healthy cross-bred heifers (Charolais, Limousin, Aberdeen Angus and Simmental) had their oestrus cycles synchronized and were inseminated with $15 \times 10^6$ sperm following a fixed-time AI protocol. The heifers were blocked by breed and age into 3 distinct treatment groups and inseminated with semen from 3 different genotype bull group each characterized by low CN (2 heifers per bull for replication), intermediate CN (2 heifers per bull), and high CN (2 heifers per bull). To maintain consistency, 2 semen straws, each derived from separate ejaculates, were used for insemination from each bull. However, a complication arose on the day of AI, necessitating an alteration in the procedure. Specifically, while 6 crossbred heifers were used for each bull genotype group; 7 crossbred heifers with Low CN, 6 crossbred heifers with Intermediate CN and 5 crossbred heifers with High CN were inseminated with frozen-thawed sperm from the respective bulls. The synchronization protocol entailed a seven-day intravaginal progesterone device (CIDR®), accompanied by intramuscular administration of gonadotrophin-releasing hormone (Ovarelin; 2 mL) at CIDR® insertion. Prostaglandin F2 alpha (Enzaprost; 5 ml) was administered intramuscularly to induce luteolysis 24 h before CIDR removal. At 54 (+/− 1) h post-CIDR® removal, heifers received a single fixed-time insemination of frozen–thawed semen. Ovarelin (2 mL) was administered intramuscularly at the time of AI. Heifers were re-evaluated after slaughter for ovulation based on the absence of the corpus luteum as well as the presence of either a large pre-ovulatory follicle or a fresh ovulation [33], and it was confirmed that all heifers were in the follicular phase of their cycle.

**Tissue collection and RNA extraction.** Twelve hours post insemination, heifers were slaughtered at a commercial abattoir. Reproductive tracts were promptly collected post-mortem, and the uterine horn on the ipsilateral side was longitudinally opened with sterile scissors. Tissue samples were acquired from the inter-caruncular region of the uterine horn near the uterine body using a sterile 8-mm biopsy punch. The endometrium was then dissected away from the myometrium. Endometrium samples were flash frozen in liquid nitrogen, transported to the laboratory, and stored at −80°C.

Total RNA isolation, RNA library preparation and sequencing was conducted by Bio-marker Technologies (BMKgene, Munster, Germany). Paired-end (150 bp) sequencing was performed on an Illumina Novaseq 6000 sequencer to a raw read depth of approximately 50 million total reads (303.20 Gb) per sample. The percentage of Q30 bases in each sample was above 98.18%.

**Quality control, mapping, and differential read count quantification.** Raw sequence reads were obtained in FASTQ format and their quality was evaluated using FastQC (v.0.12.1). The sequences from all samples were then quality-trimmed and cleared of adaptor sequences using the BBDuk Java package. For alignment, trimmed reads were semi-mapped to the bovine reference genome (ARS-UCD1.2) using Salmon [34], and uniquely mapped read counts per Ensembl (version 104) annotated gene/transcript were estimated using the Salmon–quant mode option. To ensure robustness, only genes with a minimum of five reads across all samples were considered in subsequent analyses. MultiQC analysis

was then conducted to derive essential statistics on the percentage of reads counted in the transcriptome. The Tximport function (v.1.3.9) [35] in R 4.3.1 summarized the results at the per-transcript gene level.

Differential gene expression analysis, along with data transformations and visualization, was performed using DeSeq2 (v3.18) [36] in R 4.3.1. Samples were clustered based on variance-stabilizing transformed data and visualized through principal component analysis (PCA). Differentially expressed gene (DEG) lists were created using a negative binomial generalized linear model, and comparisons were conducted among uterine biopsy groups. P-values were adjusted for multiple comparisons using the Benjamini and Hochberg (B–H) method. Genes with an adjusted P-value (False Discovery Rate (FDR)) < 0.1 were considered differentially expressed and used for further data exploration and pathway analysis. An FDR threshold of 0.1 was chosen to ensure a reasonable trade-off between discovering biologically meaningful results and limiting false positives. Gene Ontology (GO) and Kyoto Encyclopedia of Genes and Genomes (KEGG) pathway analysis were used for all DEG regardless of specific *DEFB103* CN groups. GO term was analyzed using the Gene Ontology knowledge base [37,38]. In the pathway analysis, the analysis of DEGs was performed using ClusterProfiler (v3.18.0) based on the KEGG database [39]. In this analysis, enriched GO terms and pathways were defined using a P-adjusted (FDR) ≤ 0.05.

### Recovery of sperm from bovine oviduct

Oviducts from the 18 heifers (total n = 36 paired oviducts) were collected and the ipsilateral and contralateral oviducts flushed for sperm separately. One end of each oviduct was tied with a cotton thread, spTALP solution with heparin (10 μg/mL) was prepared and injected into the oviducts with a volume of average 2 ml, and the oviducts were closed. After 60 min of incubation, the oviducts were milked, the volume assessed and sperm concentration was assessed in duplicate using a haemocytometer.

### Statistical analysis

Pearson correlation coefficients were calculated for the relationship between *DEFB103* CNV with CASA and Flow Cytometry data, and a linear regression analysis was performed between the results via GraphPad Prism (v9.5.1). All other data were checked for normality of distribution and analyzed by univariate analysis of variance (ANOVA) with Bonferroni post-hoc tests in GraphPad Prism (v 9.5.1). Statistical significance level was set at p ≤ 0.05. All results are presented as mean ± standard error of the mean.

## Results

### Impact of *DEFB103* CNV on sperm motility, mucus penetration, and oviductal epithelial binding

One objective of the study was to assess the impact of *DEFB103* CNV on sperm functional parameters, mucus penetration, and binding to the oviduct epithelium. Across all bulls, *DEFB103* CN was strongly negatively associated with total motility (p = 0.003) and progressive motility (p = 0.015) [Table 2]. Low *DEFB103* CN was associated with lower straight-line velocity (VSL; p < 0.05), average path velocity (VAP; p < 0.05), and higher sperm-oviduct binding capacity (p < 0.05; Table 3; Fig 1). Moreover, the percentage of viable sperm with high membrane fluidity in *in vitro* non-capacitating conditions demonstrated that high *DEFB103* CN was associated with higher membrane fluidity compared to the intermediate CN group (p < 0.05). There was no effect of *DEFB103* CN on the ability of sperm to penetrate artificial mucus.

**Table 2. Correlation between *DEFB103* copy number variation (CNV) and sperm functional parameters in Holstein-Friesian bulls with high (HF) and low (LF) fertility phenotypes (n = 10 per group).**

| Parameter | Pearson r | P value |
|---|---|---|
| **Computer Assisted Sperm Analysis (CASA)** | | |
| Total motility (%) | −0.64 | **0.003** |
| Progressive motility (%) | −0.54 | **0.015** |
| Curvilinear velocity (VCL; µm/s) | 0.07 | 0.76 |
| Straight-line velocity (VSL; µm/s) | 0.33 | 0.16 |
| Average path velocity (VAP; µm/s) | 0.29 | 0.21 |
| Hyperactivation (%) | −0.34 | 0.14 |
| **Flow cytometry** | | |
| Viable (%) in capacitating conditions | −0.29 | 0.21 |
| Acrosome reacted (%) in capacitating conditions | −0.21 | 0.38 |
| Acrosome intact (%) in capacitating conditions | 0.20 | 0.38 |
| High membrane fluidity (%) in capacitating conditions | −0.13 | 0.57 |
| Viable (%) in non-capacitating conditions | −0.18 | 0.44 |
| Viable, acrosome reacted (%) in non-capacitating conditions | 0.01 | 0.94 |
| Viable, acrosome intact (%) in non-capacitating conditions | −0.01 | 0.94 |
| Viable, high membrane fluidity (%) in non-capacitating conditions | −0.06 | 0.78 |

P values < 0.05 are shown in bold.

**Table 3. Effect of *DEFB103* copy number variation (CNV) on sperm functional parameters assessed using computer assisted sperm analysis (CASA), flow cytometry, mucus penetration and sperm binding to oviductal epithelial cells *in vitro* in low fertility (LF) Holstein-Friesian bulls.**

| Parameter | CNV status | | | Comparisons of each CN group (P value) | | |
|---|---|---|---|---|---|---|
| | Low | Intermediate | High | High vs Low | High vs Intermediate | Low vs Intermediate |
| **CASA** | | | | | | |
| Total motility (%) | 57.2 ± 4.39 | 50.6 ± 5.75 | 47.2 ± 4.56 | 0.82 | >0.99 | >0.99 |
| Progressive motility (%) | 42.4 ± 4.32 | 40.4 ± 5.66 | 36.9 ± 4.49 | >0.99 | >0.99 | >0.99 |
| Curvilinear velocity (VCL; µm/s) | 89.7 ± 6.70 | 107.9 ± 5.20 | 103.7 ± 9.16 | 0.42 | >0.99 | 0.21 |
| Straight-line velocity (VSL; µm/s) | 58.9 ± 4.98 | 80.7 ± 5.02 | 79.0 ± 9.49 | 0.11 | >0.99 | **0.04** |
| Average path velocity (VAP; µm/s) | 69.9 ± 4.42 | 90.7 ± 4.62 | 84.1 ± 8.67 | 0.14 | 0.58 | **0.03** |
| Hyperactivation (%) | 6.8 ± 3.31 | 3.7 ± 1.11 | 4.5 ± 2.39 | >0.99 | >0.99 | >0.99 |
| **Flow cytometer** | | | | | | |
| Viable (%) in capacitating conditions | 14.2 ± 3.90 | 10.3 ± 3.86 | 14.4 ± 3.02 | 0.6 | 0.99 | 0.57 |
| Acrosome reacted (%) in capacitating conditions | 10.1 ± 3.12 | 15.1 ± 4.10 | 5.2 ± 1.19 | 0.79 | 0.09 | 0.77 |
| Acrosome intact (%) in capacitating conditions | 89.8 ± 3.12 | 84.8 ± 4.10 | 94.8 ± 1.19 | 0.79 | 0.09 | 0.77 |
| High membrane fluidity (%) in capacitating conditions | 13.1 ± 3.91 | 5.6 ± 1.31 | 8.6 ± 1.32 | 0.29 | 0.56 | 0.051 |
| Viable (%) in non-capacitating conditions | 21.0 ± 4.11 | 16.0 ± 1.62 | 21.1 ± 2.51 | >0.99 | 0.44 | 0.47 |
| Acrosome reacted (%) in non-capacitating conditions | 1.7 ± 0.82 | 2.8 ± 1.32 | 1.0 ± 0.11 | >0.99 | 0.49 | >0.99 |
| Acrosome intact (%) in non-capacitating condition | 98.2 ± 0.85 | 97.1 ± 1.33 | 98.9 ± 0.11 | >0.99 | 0.49 | >0.99 |
| High membrane fluidity (%) in non-capacitating conditions | 3.7 ± 1.42 | 2.4 ± 12 | 5.6 ± 0.90 | 0.32 | **0.003** | 0.17 |
| **Mucus penetration (number of sperm)** | 113.5 ± 13 | 122.0 ± 59 | 122.5 ± 22.93 | 0.94 | 0.73 | 0.55 |
| **Sperm binding to oviductal epithelium (per 0.1 mm²)** | 1.6 ± 0.18 | 1.0 ± 0.08 | 1.1 ± 0.04 | 0.051 | 0.97 | **0.03** |

Values are mean ± s.e.m. n = 3 bulls per group. P values < 0.05 are shown in bold.

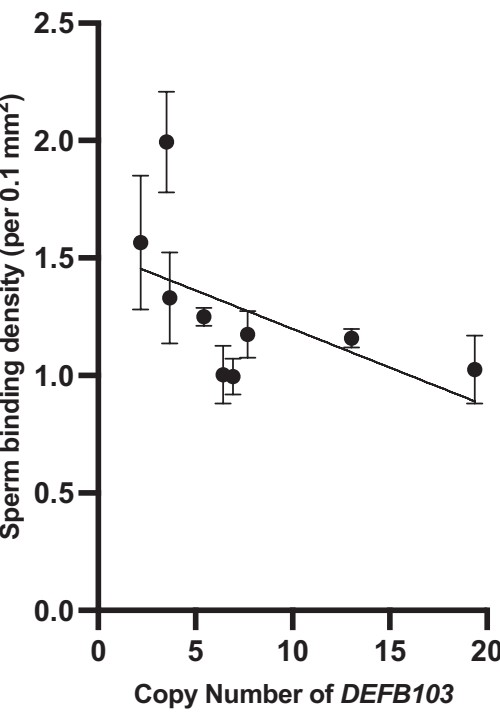

**Fig 1. The ability of frozen-thawed sperm from low fertility Holstein-Friesian bulls with varying *DEFB103* copy number variation (CNV) to bind to the oviductal epithelial *in vitro*.** Mean and standard errors of three replicates are shown for low fertility bulls (n = 9). The central slope indicates the regression curve obtained through linear regression analysis. r = −0.5, p = 0.006.

## Effect of *DEFB103* copy number variation on sperm migration to the oviducts and the uterine transcriptomic response

Given the effect of low CNV on sperm phenotype, in particular on sperm oviductal binding, the effect of the sperm from bulls with different *DEFB103* CN on sperm migration to the oviducts and uterine response *in vivo* was investigated. The same number of sperm were inseminated into each heifer ($15 \times 10^6$) and an average of $0.16 \times 10^6$ sperm were recovered from the contralateral oviduct and $0.34 \times 10^6$ sperm were recovered from the ipsilateral oviduct. There was no difference (p > 0.05) in the number of sperm recovered from the ipsilateral and contralateral oviducts and a large variation between individual animals was observed. Similarly, no relationship between sperm number recovered by flushing and the *DEFB103* CN of the bull was detected (Fig 2).

Principal component analysis of global transcript levels from the 18 different heifers uterine tissue showed no clear clustering based on the *DEFB103* CN of the bull whose semen was used for insemination (S1 Fig) indicating an absence of global differences due to sperm treatment. This is likely accounted for by the complexity of diverse sperm-uterine interactions occurring simultaneously as well as the heterogeneity due to the use of different heifers. To identify more subtle differences between treatments, three pairwise comparisons were performed: 1. between heifers inseminated with semen from high *DEFB103* CN bulls and those inseminated by low CN bulls; 2. between heifers inseminated with semen from high *DEFB103* CN bulls and those inseminated by intermediate CN bulls; and 3. between heifers inseminated with semen from intermediate *DEFB103* CN bulls and those inseminated by low CN bulls.

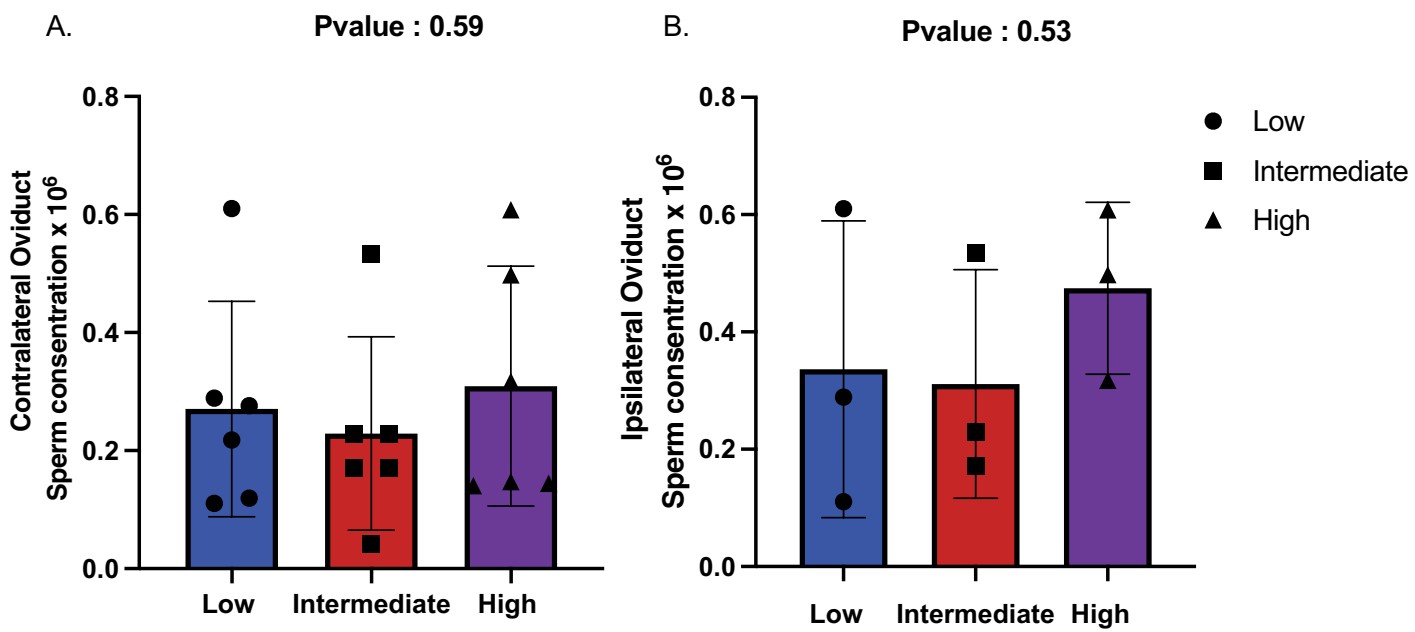

**Fig 2. The ability of frozen-thawed sperm from low fertility Holstein-Friesian bulls (n = 3 bulls per group) with varying *DEFB103* copy number to populate the oviduct *in vivo*.** (A) Sperm recovered from the contralateral oviduct, and (B) Sperm recovered from the ipsilateral oviduct after flushing. The X-axis represents the genotype, while the Y-axis indicates the sperm concentration (x $10^6$).

A total of 58 genes were differentially expressed in the uterus post-insemination using sperm from *DEFB103* bulls using a threshold of FDR < 0.1 (Full list shown in S1 Table). Some of these genes had a low $\log_2$ fold change (FC) and those with a $\log_2$FC ≤ −2 or are shown in Table 4. In comparison 1, the high CN group showed increased expression of 5 genes (*KYAT3, CADM3, SEMA3C, SEMA3A,* and *SLC35F1*) compared to the low CN group, while the expression of 6 genes (*SKAP2, RPS3A, TGIF1, EFR3B,* and *CYP2C87*), including Calponin 3 (*CNN3*), was decreased in the same comparison.

The greatest difference in transcriptomic response was detected when comparing uterine responses to sperm from high and intermediate CN bulls (41 genes $\log_2$FC > 2, Full list shown in S1 Table). There was a decrease in the expression of 12 genes, especially *RPS3A, RPS4, AKR1C1, GRO1, CXCL3,* and *ITLN2,* in the high CN group compared to the intermediate CN group. In contrast, among the up-regulated genes, *BRINP3, PRRT3, GABRA4,* and *CDH9* showed the largest increases in their expression levels in response to sperm in the uterine endometrium. In comparison 3, the intermediate CN group was compared with the low CN group, a total of 6 genes were differentially expressed. Among these, *CNN3* showed an almost 8.5-fold lower expression, while *GRO1, CXCL3,* and *RPS4* showed significantly higher expression.

Gene ontology analysis of DEG, identified in the three pairwise comparisons, indicated cell migration as a key process influenced by *DEFB103* CN of the bull (Table 5). Kyoto Encyclopedia of Genes and Genomes (KEGG) analysis identified several potential pathways affected – of particular interest was cell adhesion processes and the *NFKBIA/GRO1/CXCL3* signalling axis (Table 6). Low *DEFB103* CN in bulls was associated with increased expression of genes promoting uterine preparation for embryonic development and chemotaxis. Conversely, high *DEFB103* CN was associated with the high expression of genes affecting cell adhesion and proliferation, and with low expression of genes that are crucial for protein synthesis and

**Table 4. Differential gene expression analysis results comparing intermediate copy number (CN) bulls versus low CN bulls, low CN bulls versus high CN bulls and intermediate CN bulls versus high CN bulls.**

| Gene Symbol | Description | Gene ID | Log₂ FC | FDR |
|---|---|---|---|---|
| **Sperm from High CN versus Low CN bulls** | | | | |
| CNN3 | Calponin 3 | ENSBTAG00000024407 | −9 | $6.70 \times 10^{-02}$ |
| RPS3A | Ribosomal Protein S3A | ENSBTAG00000006383 | −8.46 | $4.10 \times 10^{-02}$ |
| CYP2C87 | Cytochrome P450, family 2, subfamily C, polypeptide 87 | ENSBTAG00000037795 | −2.6 | $9.00 \times 10^{-02}$ |
| **Sperm from High CN bulls versus Intermediate CN bulls** | | | | |
| RPS3A | Ribosomal Protein S3A | ENSBTAG00000006383 | −9.18 | $3.80 \times 10^{-03}$ |
| BRINP3 | BMP/retinoic acid inducible neural specific 3 | ENSBTAG00000008420 | 5.53 | $8.10 \times 10^{-02}$ |
| CDH9 | cadherin 9 | ENSBTAG00000018234 | 7.46 | $1.20 \times 10^{-03}$ |
| RPS4 | Small ribosomal subunit protein | ENSBTAG00000031205 | −7.86 | $3.30 \times 10^{-03}$ |
| ITLN2 | intelectin 2 | ENSBTAG00000048662 | −6.14 | $4.90 \times 10^{-02}$ |
| GABRA4 | Gamma-aminobutyric acid type A receptor subunit alpha4 | ENSBTAG00000016645 | 4.78 | $6.30 \times 10^{-02}$ |
| PRRT3 | Proline rich transmembrane protein 3 | ENSBTAG00000010836 | 3.08 | $1.30 \times 10^{-02}$ |
| SAA | Serum amyloid A protein | ENSBTAG00000022395 | −2.04 | $4.90 \times 10^{-02}$ |
| AKR1C1 | Aldo-keto reductase family 1, member C1-like | ENSBTAG00000037509 | −2.69 | $3.00 \times 10^{-02}$ |
| GRO1 | Chemokine (C-X-C motif) ligand 1 (melanoma growth stimulating activity, alpha) | ENSBTAG00000037558 | −3.55 | $6.90 \times 10^{-02}$ |
| CXCL3 | Chemokine (C-X-C motif) ligand 3 | ENSBTAG00000037778 | −2.39 | $6.90 \times 10^{-02}$ |
| NRXN1 | Neurexin 1 | ENSBTAG00000046199 | 2.11 | $5.70 \times 10^{-02}$ |
| **Sperm from Intermediate CN bulls versus Low CN bulls** | | | | |
| RPS4 | Small ribosomal subunit protein | ENSBTAG00000031205 | 8.58 | $1.70 \times 10^{-04}$ |
| CNN3 | Calponin 3 | ENSBTAG00000024407 | −8.47 | $6.50 \times 10^{-02}$ |
| GRO1 | Chemokine (C-X-C motif) ligand 1 (melanoma growth stimulating activity, alpha) | ENSBTAG00000037558 | 3.67 | $6.50 \times 10^{-02}$ |
| CXCL3 | Chemokine (C-X-C motif) ligand 3 | ENSBTAG00000037778 | 2.54 | $6.50 \times 10^{-02}$ |

Genes with statistically significant differences in expression (FDR < 0.1) and log₂ fold changes < −2 or >+2 were filtered for inclusion in the table. FC refers to Fold Change and FDR refers to False Discovery Rate.

**Table 5. Gene Ontology (GO) analysis for heifers inseminated with frozen-thawed semen from low fertility bulls, categorized by different *DEFB103* genotypes.**

**GO: Biological process**

| Category | GO ID | Gene | Definition | P-value |
|---|---|---|---|---|
| Neural crest cell migration | GO:0001755 | SOX8/TWIST1/ SEMA3C/SEMA3A | The characteristic movement of cells from the dorsal ridge of the neural tube to a variety of locations in a vertebrate embryo | $8 \times 10^{-03}$ |
| Mesenchymal cell migration | GO:0090497 | SOX8/TWIST1/ SEMA3C/SEMA3A | The orderly movement of a mesenchymal cell from one site to another, often during the development of a multicellular organism | $6 \times 10^{-03}$ |

This analysis includes all differentially expressed genes (DEGs) with a corrected p-value of less than 0.01.

**Table 6. Analysis of the top 5 significant KEGG pathways for heifers inseminated with frozen-thawed semen from low fertility bulls, categorized by different *DEFB103* genotypes.**

| KEGG ID | Category | Description | Gene in pathway | P-value |
|---|---|---|---|---|
| bta04514 | Signalling molecules and interaction | Cell adhesion molecules | CADM3/LRRC4B/NRXN1/BOLA | $7.20 \times 10^{-04}$ |
| bta04360 | Development and regeneration | Axon guidance | ROBO1/ROBO2/SEMA3A/SEMA3C | $1.02 \times 10^{-03}$ |
| bta04657 | Immune system | IL-17 signalling pathway | NFKBIA/GRO1/CXCL3 | $1.60 \times 10^{-03}$ |
| bta04064 | Signal transduction | NF-kappa B signalling pathway | NFKBIA/GRO1/CXCL3 | $2.60 \times 10^{-03}$ |
| bta04668 | Signal transduction | TNF signalling pathway | NFKBIA/GRO1/CXCL3 | $3.90 \times 10^{-03}$ |

This analysis includes all differentially expressed genes (DEGs) with a corrected p-value of less than 0.01.

immune response, potentially affecting these processes in the uterine environment (all details shown in S1 Table).

## Discussion

In this study, we investigated the effect of bull *DEFB103* CNV on sperm motility, kinematic and functional parameters in Holstein-Friesian bulls. From an extensive panel of field fertility ranked bulls used in AI, 20 High Fertility (HF) and Low Fertility (LF) bulls were selected for CNV functional analysis. *DEFB103* CN was strongly negatively associated with total motility and progressive motility (−0.6 and −0.5, respectively), potentially explaining a proportion of the variation in fertility between these groups. This builds on previous work showing that *DEFB103* is expressed in the caput epididymis, and where extensive multiallelic CNV negatively correlates with its expression level in the testis and it is upregulated at sexual maturity [26]. Consequently, reduced expression of *DEFB103* protein may impair sperm motility and function, which may explain the lower progressive and overall motility observed in bulls with higher *DEFB103* CN.

CNV typing showed the presence of high CNV in three bulls within the LF bull group only and therefore due to inadequate CNV diversity in HF bulls, the remainder of the study specifically examined sperm progression to the oviducts *in vivo* and the uterine transcriptomic response to sperm in LF bulls only. Despite the deliberate omission of HF bulls, this investigation remains relevant as significant variation in field fertility exists within the LF bull group (varying from −12.3 to −3 on the adjusted sire fertility model) and the research question assessed the contribution of CNV to this variation. The HF bulls were not included in the *in vivo* experiment as there was insufficient CNV. Within the LF group, a statistically significant effect on two critical sperm kinematic parameters - VSL and VAP, with sperm from low CV bulls displaying reduced velocity. The observed disparities in sperm kinematic parameters suggest that high or low fertility cannot be attributed solely to *DEFB103* CN and it is possible that an additive effect across the β-defensin gene locus could have a stronger effect on the overall fertility phenotype.

Upon entering the female reproductive tract, sperm undergo crucial membrane transformations that are essential for fertilization [40] and previous research showed that LF bull sperm had reduced membrane fluidity under *in-vitro* capacitating conditions compared to HF bulls [29]. Results from the current study suggests that high *DEFB103* increases membrane fluidity, especially under capacitating conditions, which may facilitate stronger binding to the oviductal epithelium. A number of studies have also shown differences in the ability of sperm to bind to the oviductal epithelium between HF and LF bulls [11]. Considering β-defensins are known to play a critical role in this process [20], results from this study illustrated that high *DEFB103* CN reduced the oviductal binding of sperm suggesting that lower *DEFB103* CN might improve fertility outcomes by enhancing sperm binding capabilities.

After demonstrating a clear effect of *DEFB103* indirectly on sperm function, we sought to determine whether the uterine response varied depending on the sperm from bulls with different *DEFB103* CN. Overall, transcriptomic differences were modest, but *DEFB103* CN had significant effects on expression of a limited number of genes with roles in sperm migration, immune signalling, and chemotaxis pathways. *RPS3A*, which is an integral part of ribosomal proteins, is involved in actin cytoskeleton regulation, cell adhesion, and tissue remodelling [41], and showed 8.46- and 9.18-$\log_2$FC decreases in expression levels in the High CN group compared to the Low and Intermediate CN groups, respectively, which may impact fertility-related protein translation. An almost 9 $\log_2$FC increase in the expression

level of Calponin 3 (*CNN3)* was detected in the Low CN group compared to others. *CNN3* regulates epithelial contractile activity by localizing to actin filaments, and interestingly, studies have shown that this is essential for embryonic development and viability in mice [42]. More importantly, *CNN3*, which showed increased expression in the low *DEFB103* CN group, has been reported to be involved in the cytoskeletal rearrangement required for trophoblastic fusion in humans, a critical process for embryo implantation and interaction with the decidualized maternal uterus [43]. *RPS4*, ribosomal protein, along with two chemokines, *GRO1* and *CXCL3*, were significantly upregulated in response to sperm from intermediate CN bulls compared to sperm from the low CN group. *RPS4* plays a role in spermatogenesis [44], suggesting that significant changes in intercellular communication, cell cycle regulation, and protein synthesis occur. *GRO1* has previously been shown to have high expression levels in decidual endometrial stromal cells in response to products secreted by trophoblasts in humans [45]. *CXCL3*, has been shown to play a role in the migration, invasion, proliferation, and tube formation of trophoblast cells [46]. Taken together, these findings suggest that *DEFB103* may contribute to chemotaxis during sperm motility and settlement within the uterus. In contrast, Intelectin 2 (*ITLN2*) expression was 6.14 $\log_2$FC lower in the high CN group compared to the intermediate CN group. *ITLN2*, plays a role in the innate immune response [47], and may influence immune responses or tissue repair processes in the uterine environment due to its reduced expression in sperm from high CN bulls. There was an increase in Cadherin 9 (*CDH9*) and Gamma-aminobutyric acid receptor subunit alpha-4 (*GABRA4*) expression in sperm from high CN bulls versus intermediate CN bull group. Since *CDH9* plays a role in cell-cell adhesion, its increased expression in sperm may increase the interaction between sperm and the uterine epithelium. While this could potentially improve sperm retention and survival in the female reproductive tract; increased expression level of *GABRA4* may negatively affect the preparation of the uterine endometrium for implantation. It has previously been reported that *GABA* and synthetic *GABA* receptor ligands in mice can negatively affect preimplantation embryos through their receptors [48].

## Conclusion

*DEFB103* CNV exerts a significant impact on LF bulls by affecting various sperm functional parameters and eliciting distinct transcriptomic responses in the uterus post-insemination. Lower *DEFB103* CN, associated with higher *DEFB103* expression levels, likely enhances sperm motility and oviductal epithelium binding capacity by increasing protein availability and affecting charge-mediated progression to the site of fertilisation. Additionally, *DEFB103* CN may alter the uterine immune response to sperm from bulls with low-CN, and therefore affecting early pregnancy establishment although further conclusive investigation is required. These promising findings illustrate the likely impact of *DEFB103* CNV on multiple aspects of bovine reproduction and suggests that further dissection of the fertility roles of β-defensins in a larger panel of cattle and breeds is now warranted.

## Supplementary information

**S1 Fig. Principal Component Analysis (PCA) showing the distribution of RNA-seq results.**
(PDF)

**S1 Table. Differential gene expression results including GO and KEGG analysis.**
(XLSX)

## Acknowledgements

The authors acknowledge Dr Ciara O' Meara from the National Cattle Breeding Centre and Mr Enda Dooley from Dovea Genetics for provision of semen straws. We are also grateful to Dr Miriama Stiavnicka for providing CASA and Flow cytometer data for the HF and LF phenotype bulls. We thanks to Kaitlyn Weldon and Nathan Johnston from University of Limerick for help to collect the female reproductive tract samples from the commercial abattoir. This research used the ALICE High Performance Computing facility at the University of Leicester.

## Author contributions

**Conceptualization:** Ozge Sidekli, Edward J. Hollox, Sean Fair, Kieran G. Meade.

**Data curation:** Ozge Sidekli.

**Formal analysis:** Ozge Sidekli, Sean Fair.

**Funding acquisition:** Ozge Sidekli, Edward J. Hollox.

**Investigation:** Ozge Sidekli, Edward J. Hollox, Sean Fair, Kieran G. Meade.

**Methodology:** Ozge Sidekli, Edward J. Hollox.

**Project administration:** Edward J. Hollox, Kieran G. Meade.

**Supervision:** Edward J. Hollox, Sean Fair, Kieran G. Meade.

**Writing – original draft:** Ozge Sidekli, Kieran G. Meade.

**Writing – review & editing:** Ozge Sidekli, Edward J. Hollox, Sean Fair, Kieran G. Meade.

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
