## [Decision Letter · Decision Letter 0]

15 Dec 2024

PONE-D-24-51902Impact of β-defensin 103 (DEFB103) copy number variation on bull sperm parameters and post-insemination uterine gene expressionPLOS ONE

Dear Dr. Meade,

Thank you for submitting your manuscript to PLOS ONE. After careful consideration, we feel that it has merit but does not fully meet PLOS ONE’s publication criteria as it currently stands. Therefore, we invite you to submit a revised version of the manuscript that addresses the points raised during the review process.

**ACADEMIC EDITOR: Please respond carefully for all reviewers comments. **

We look forward to receiving your revised manuscript.

Kind regards,

Ayman A. Swelum

Academic Editor

PLOS ONE

Journal Requirements:

OS was funded by the Turkish Ministry of National Education, Republic of Turkiye postgraduate study abroad program. Support for travel for OS was provided in part by the Genetics Society. SF was funded by the Marie Skłodowska-Curie Doctoral Network, BullNet, grant number 101120104.   

Reviewers' comments:

Reviewer's Responses to Questions

**Comments to the Author**

1. Is the manuscript technically sound, and do the data support the conclusions?

Reviewer #1: Yes

Reviewer #2: No

Reviewer #3: Yes

Reviewer #4: Yes

2. Has the statistical analysis been performed appropriately and rigorously? 

Reviewer #1: Yes

Reviewer #2: No

Reviewer #3: Yes

Reviewer #4: No

3. Have the authors made all data underlying the findings in their manuscript fully available?

Reviewer #1: Yes

Reviewer #2: Yes

Reviewer #3: Yes

Reviewer #4: Yes

4. Is the manuscript presented in an intelligible fashion and written in standard English?

Reviewer #1: Yes

Reviewer #2: No

Reviewer #3: Yes

Reviewer #4: Yes

5. Review Comments to the Author

Reviewer #1: This is a very good study and was intelligently written. However, there are minor corrections to be made. The authors did infer but did not categorically state the justification for the study, please give a statement to justify the study at the end of the introduction. The pages of the document need to be numbered and authors should ensure consistency in the use of 'et al'. Other comments are highlighted in the document attached.

Reviewer #2: General Comments

The manuscript PONE-D-24-51902, titled "Impact of β-defensin 103 (DEFB103) copy number variation on bull sperm parameters and post-insemination uterine gene expression" addresses the potential impact of DEFB103 copy number variation (CNV) on bull fertility traits, including sperm functionality and uterine transcriptomic responses. While the topic is interesting and potentially significant for livestock breeding programs, the manuscript has several critical issues that limit its clarity, scientific rigour, and impact.

Major Concerns

1. Abstract

The abstract focuses heavily on presenting results while providing minimal information about the methodology, making it unbalanced and less informative. Its disjointed structure lacks a smooth flow, resulting in a "jerky" reading experience. The overgeneralization of findings further detracts from its clarity and precision.

Recommendation: A better balance between background, methods, results, and implications is needed to enhance its effectiveness.

2. Unclear Objectives

The objectives of the study are not clearly stated. While the manuscript appears to explore the effects of DEFB103 CNV on fertility traits, the absence of a concise hypothesis or research question makes it challenging to understand the study's scope and rationale.

Recommendation: Clearly articulate the study objectives and hypothesis in the introduction, ensuring alignment with the methods and results.

3. Introduction: Lack of Focus

The introduction includes extraneous information, such as discussions on cow fertility and seasonal calving systems, which are not directly relevant to the study's focus on bull fertility. Additionally, the rationale for studying DEFB103 CNV is underdeveloped, and the study's novelty is unclear.

Recommendation: Streamline the introduction to focus on:

a. The importance of bull fertility for livestock productivity.

b. Knowledge gaps regarding the role of β-defensins in reproduction.

c. The specific rationale for investigating DEFB103 CNV in bull fertility.

4. Study Design and Methodological Issues

• Sample Size: The small sample size (20 bulls and 18 heifers) limits the statistical power and generalizability of the findings. Additionally, there is no clear explanation of how these bulls were selected from the initially mentioned 840 bulls.

• Methodological Inconsistencies: The decision to focus only on low-fertility bulls for transcriptomic analysis is not well justified, and the study does not adequately control for confounding factors. Moreover, reference no. 28 is incorrectly cited in the methodology section. In actual study cited as reference no.28 is totally opposite than what authors have perceived. It was difficult to understand "why authors decided to use crossbred heifers for Holstein bulls". There too many dependent variables studied on very limited sample size.

Recommendation: Increase the sample size and justify the experimental design more vigorously. Address potential confounders comprehensively.

5. Data Interpretation and Overstatement of Findings

The findings, particularly in the transcriptomic analysis, are overstated, given the modest differential expression results (58 genes, FDR < 0.1). Many conclusions lack mechanistic explanations or clear connections to fertility outcomes.

Recommendation: Interpret results more cautiously, emphasizing biologically meaningful findings and acknowledging limitations.

6. Overlap with Previously Published Studies

The manuscript shows substantial overlap with previously published work by the same authors, particularly Sidekli et al. (2024). While including transcriptomic analysis adds some novelty, the manuscript does not adequately differentiate its contributions from prior publications.

Recommendation: Delineate how this study builds upon or diverges from earlier work. Emphasize novel aspects and ensure proper acknowledgement of prior findings.

7. Discussion and Conclusion

The discussion overstates the significance of findings and lacks integration with existing knowledge on β-defensins, making speculative claims without sufficient data support. It fails to address critical limitations, including the small sample size and lack of estrous cycle control. The conclusion is overly broad, misaligning with the study's modest results and lacking clear practical implications or future research directions.

Recommendation: Reframe the discussion and conclusion to provide balanced interpretations, acknowledge limitations, and suggest realistic next steps.

8. Writing Style and Presentation

The writing is dense, and key points are difficult to extract due to overly technical language and a lack of logical flow. Figures and tables lack clarity in presenting results; some are not adequately described in the text. Additionally, the absence of line numbers up to page 22 significantly complicated the review process.

Recommendation: Simplify the language and improve the manuscript's organization for better readability. Revise figures and tables to enhance clarity and alignment with the text. Include line numbers throughout the manuscript for ease of review.

Minor Comments

• References: Replace tangential or outdated references with more recent and directly relevant studies.

Reviewer #3: Reviewer’s comments on manuscript ID PONE-D-24-51902

This manuscript entitled “Impact of b-defensin 103 (DEFB103) copy number variation on bull sperm parameters and post-insemination uterine gene expression” and submitted by Kieran Meade investigate the possible role of DEFB103 as a potential marker of fertility in bovine. Firstly, the authors examine the DEFB103 copy number (CN) in spermatozoa of bulls classified as low or high fertility, followed by investigating various parameters of sperm derived from low fertile bulls with known CN values (low, intermediate, and high). Additionally, the authors examine the differential impacts on uterine tissue gene expressions 12h post-insemination.

Overall, the manuscript is well-written. It addresses a subject of interest to better contribute to selecting dairy cows. The authors provide sufficient background information to understand the study's rationale and cite the most appropriate references to support the study goal. The experimental procedures are well described, and relevant techniques are explained. However, the high CN diversity among high-fertility bulls appears to have excluded this group from the remaining study. I believe the complete omission of including high fertile bulls in the second part of the study may have removed the most crucial information related to the abundance of CN in low vs high fertile bulls. This leaves the reader asking how the results obtained with low fertile bulls in the second study will be significant. Regardless, the findings are appropriately discussed within the context of the current literature. In my opinion, the absence of high fertile bull, at least a comparison between low CN values, is a low point of the study, which will remain of interest to those in the closely related field of science.

Additional comments are listed below.

Specific comments

Table 1. Could the authors add new lines after H10 and L10, providing the average(±sd/sem) of all values listed above? This will allow for a fast grasp of the information related to comparing H vs L.

Could the authors provide an image of the artificial mucus they made? This would be of interest to those desiring to replicate it.

A clarification is needed to prepare the explants: “These were cultured in M199: - what was the final volume of the co-incubated explants and sperm? – Why were the 75 ul taken for? Counting loose sperm or something else.

Other clarification is needed for the heifer breed selection. Four breeds are mentioned, but only two were used per individual bull of low, intermediate, or high CN. Could this mean that each CN value group included all four heifer breeds, with two bred with the single bull? How do the authors end up with 18 heifers? - Would two heifers per bull be enough for complete reliance to generate reliable data?

Overall, the absence of page and line numbers makes a thorough review with specific comments difficult.

Reviewer #4: The study explores the relationship between copy number variation (CNV) in the β-defensin 103 (DEFB103) gene and various aspects of bull fertility, including sperm motility, binding, and uterine gene expression post-insemination. Using a combination of in vitro and in vivo methodologies, the authors investigate molecular and physiological effects of DEFB103 CNV on fertility outcomes in Holstein-Friesian bulls. The findings suggest that lower DEFB103 CN is associated with improved sperm motility and enhanced uterine responses, providing a novel genetic marker for fertility assessment in cattle.

A. Abstract and Introduction

1. Abstract: The abstract is slightly dense. Emphasizing key findings would improve accessibility for a broader audience.

2. Introduction: The link between β-defensin functionality and fertility could be expanded with more emphasis on prior studies involving other β-defensin genes in cattle.

B. Methods

1. Bull Selection: While the selection criteria for bulls are explained, providing more details about how "low" and "high" fertility bulls were identified from field fertility data would add clarity.

2. Statistical Analysis: Explain the rationale for choosing an FDR threshold of 0.1 for differential gene expression.

3. Sample Size Justification: The study relies on a small sample size (e.g., n=3 per CNV class for some analyses). A discussion of the limitations posed by this and how they were mitigated would be valuable.

C. Results and Discussion

1. Figure 1 (binding to oviductal epithelium) could benefit from clearer labeling and legends explaining the significance of observed trends.

2. While the discussion addresses findings in-depth, it occasionally becomes speculative (e.g., roles of CNN3 and RPS3A in embryo implantation). Stronger links to experimental data are needed to support such claims.

3. Comparative discussion of findings with other β-defensin studies across species could highlight the broader relevance of the results.

4. Terminology Consistency: Terms like "low fertility" and "low DEFB103 CN" need consistent usage throughout to avoid confusion.

5. Data Accessibility: RNA-seq data availability is noted, but providing a direct repository link in the main text or supplementary information would be helpful for replication purposes.

4. Broader Impacts: This study offers valuable insights for livestock genetics and fertility management. While the findings are preliminary, the implications for breeding strategies and identifying molecular markers for fertility are significant. Further research with larger sample sizes and across diverse breeds would strengthen these findings.

6. PLOS authors have the option to publish the peer review history of their article (what does this mean? ). If published, this will include your full peer review and any attached files.

**Do you want your identity to be public for this peer review?** For information about this choice, including consent withdrawal, please see our Privacy Policy .

Reviewer #1: No

Reviewer #2: **Yes: ** Khalid Mahmood

Reviewer #3: **Yes: ** Jean M Feugang

Reviewer #4: No

---

## [Author Response · Author response to Decision Letter 0]

3 Jan 2025

Please see attached rebuttal letter

---

## [Decision Letter · Decision Letter 1]

12 Jan 2025

PONE-D-24-51902R1Impact of β-defensin 103 (DEFB103) copy number variation on bull sperm parameters and post-insemination uterine gene expressionPLOS ONE

Dear Dr. Meade,

Thank you for submitting your manuscript to PLOS ONE. After careful consideration, we feel that it has merit but does not fully meet PLOS ONE’s publication criteria as it currently stands. Therefore, we invite you to submit a revised version of the manuscript that addresses the points raised during the review process.

**ACADEMIC EDITOR: The manuscript was improved greatly. However, minor revision is needed before its acceptance. **

We look forward to receiving your revised manuscript.

Kind regards,

Ayman A. Swelum

Academic Editor

PLOS ONE

Journal Requirements:

Reviewers' comments:

Reviewer's Responses to Questions

**Comments to the Author**

1. If the authors have adequately addressed your comments raised in a previous round of review and you feel that this manuscript is now acceptable for publication, you may indicate that here to bypass the “Comments to the Author” section, enter your conflict of interest statement in the “Confidential to Editor” section, and submit your "Accept" recommendation.

Reviewer #2: (No Response)

Reviewer #4: All comments have been addressed

2. Is the manuscript technically sound, and do the data support the conclusions?

Reviewer #2: Yes

Reviewer #4: Yes

3. Has the statistical analysis been performed appropriately and rigorously? 

Reviewer #2: Yes

Reviewer #4: Yes

4. Have the authors made all data underlying the findings in their manuscript fully available?

Reviewer #2: Yes

Reviewer #4: Yes

5. Is the manuscript presented in an intelligible fashion and written in standard English?

Reviewer #2: Yes

Reviewer #4: Yes

6. Review Comments to the Author

Reviewer #2: The manuscript has markedly improved in clarity and rigor. The revised version effectively addresses the initial review concerns, clearly presenting the methodologies used. I commend the authors for their thorough responses and the significant revisions made.

However, there are a few areas where minor textual clarifications could enhance readability and precision. These suggestions are detailed in the comment boxes of the uploaded file.

Overall, the manuscript now offers a valuable contribution to the literature on β-defensin 103 and its impact on bull sperm parameters and post-insemination uterine gene expression.

Reviewer #4: The authors have revised the manuscript and addressed all of my comments. It is recommended for publication.

7. PLOS authors have the option to publish the peer review history of their article (what does this mean? ). If published, this will include your full peer review and any attached files.

**Do you want your identity to be public for this peer review?** For information about this choice, including consent withdrawal, please see our Privacy Policy .

Reviewer #2: **Yes: ** Khalid Mahmood

Reviewer #4: No

---

## [Editor Report · Decision Letter 2]

30 Jan 2025

Impact of β-defensin 103 (DEFB103) copy number variation on bull sperm parameters and post-insemination uterine gene expression

PONE-D-24-51902R2

Dear Dr. Meade,

We’re pleased to inform you that your manuscript has been judged scientifically suitable for publication and will be formally accepted for publication once it meets all outstanding technical requirements.

Kind regards,

Ayman A Swelum

Academic Editor

PLOS ONE
---

## [Editor Report · Acceptance letter]

PONE-D-24-51902R2

PLOS ONE

Dear Dr. Meade,

I'm pleased to inform you that your manuscript has been deemed suitable for publication in PLOS ONE. Congratulations! Your manuscript is now being handed over to our production team.

Kind regards,

on behalf of

Professor Ayman A Swelum

Academic Editor

PLOS ONE